DOI: 10.1038/s41467-018-03996-1　　**OPEN**

# Dinosaur diversification linked with the Carnian Pluvial Episode

Massimo Bernardi [1,2], Piero Gianolla [3], Fabio Massimo Petti [1,4], Paolo Mietto[5] & Michael J. Benton [2]

Dinosaurs diversified in two steps during the Triassic. They originated about 245 Ma, during the recovery from the Permian-Triassic mass extinction, and then remained insignificant until they exploded in diversity and ecological importance during the Late Triassic. Hitherto, this Late Triassic explosion was poorly constrained and poorly dated. Here we provide evidence that it followed the Carnian Pluvial Episode (CPE), dated to 234–232 Ma, a time when climates switched from arid to humid and back to arid again. Our evidence comes from a combined analysis of skeletal evidence and footprint occurrences, and especially from the exquisitely dated ichnofaunas of the Italian Dolomites. These provide evidence of tetrapod faunal compositions through the Carnian and Norian, and show that dinosaur footprints appear exactly at the time of the CPE. We argue then that dinosaurs diversified explosively in the mid Carnian, at a time of major climate and floral change and the extinction of key herbivores, which the dinosaurs opportunistically replaced.

[1] MUSE—Museo delle Scienze, Corso del Lavoro e della Scienza 3, 38122 Trento, Italy. [2] School of Earth Sciences, University of Bristol, Bristol BS8 1RJ, UK. [3] Dipartimento di Fisica e Scienze della Terra, Università di Ferrara, via Saragat 1, 44100 Ferrara, Italy. [4] PaleoFactory, Dipartimento di Scienze della Terra, Sapienza Università di Roma, Piazzale Aldo Moro, 5, 00185 Rome, Italy. [5] Dipartimento di Geoscienze, Università degli studi di Padova, via Gradenigo 6, I-35131 Padova, Italy. Correspondence and requests for materials should be addressed to M.B. (email: massimo.bernardi@muse.it)

Dinosaurs are exemplars of an evolutionarily successful group[1]. The clade Dinosauria includes the large Mesozoic reptiles, as well as birds, comprising more than 11,000 species, with extremely disparate morphologies. The dinosaur stem lineage originated in the early Triassic, in the maelstrom of perturbed environments and recovering faunas following the devastation of the Permian-Triassic mass extinction[2]. However, the mode and timing of the origin and diversification of the dinosaurs have so far been unresolved.

The clade Dinosauria rose to its ecological acme through the 50 myr of the Triassic, and two key steps can be recognised: the origin of Dinosauria in the early-Middle Triassic, followed by a span of some 20 myr during which dinosaurs existed at low diversity, and then the explosive diversification of the clade in the early part of the Late Triassic[2]. The focus here is on this latter diversification event, when new lineages emerged and the ecological dominance of the group, measured by relative abundance of specimens in well documented faunas, shifted from <5 to >90%[2–4]. The diversification of dinosaurs at this point, especially herbivorous forms, followed extinctions of previously ecologically significant herbivorous groups such as dicynodonts and rhynchosaurs.

Over the years, this explosion, which we term the 'dinosaur diversification event' (DDE), was dated variously to the Carnian and Norian, depending on continuing revision of continental stratigraphy[5,6]. The problem has been that there was no independent system of dating for the bulk of Triassic terrestrial rock successions. Initially, they were roughly aligned in time according to their contained tetrapod faunas[7], providing approximate early, middle, and late Triassic divisions. However, the late Triassic is over 35 myr long, and a great deal happened during that time, meaning that a reliable way to subdivide this long interval is required.

This uncertainty highlights the need to avoid circularity in dating the key events in the origin of dinosaurs by not using tetrapod faunas to provide the dating. We believe that we have the solution here, which is to use rock sections that are dated independently of the tetrapod fossils, by means of magnetostratigraphy, radioisotopic dates, and correlations to marine successions dated by ammonites, conodonts, and other fossils. Recent attempts to understand the relative ordering of Triassic terrestrial tetrapod faunas[2–4,8–10] were founded in part on magnetostratigraphic data that provided correlations between red bed successions of Europe and North America[11,12], combined with some exact age dates in the Carnian, Norian, and Rhaetian[13], some of them directly from tetrapod-bearing sedimentary units. Here and there, cross-overs between terrestrial and marine sediments, for example around the edges of the Germanic Basin, provided additional independent age tie-points. Nonetheless, critical faunas with early dinosaurs from the North American south-west, India and, at least in part, Argentina and Brazil, currently lack precise independent age dating, and so cannot be used to test the timing of the event.

There was an important climate-change event in the Carnian, the Carnian Pluvial Episode (CPE)[14], termed also "Carnian Humid Episode", "Carnian Pluvial Event" or "Wet Intermezzo"[15] that corresponds to a time of major turnover in the oceans and on land. Could this switch from arid to humid and then back to arid conditions mark the trigger for the replacement of precursor herbivorous reptiles by the dinosaurs[3,8]?

The Permian and Triassic sedimentary sequences of the Alpine region offer a unique opportunity in this respect. The palaeo-geographic setting of the Dolomites and nearby areas (NE Italy), gave rise to a unique geological situation, now well exposed in several sections, in which marine sediments, continental deposits, and volcanites interfinger[16]. Correlation of sections in the region, using marine fossils, magnetostratigraphy, and exact radio-isotopic age dates, has allowed the development of a precise framework of biostratigraphic and chronological data, which can now be used to date faunas and events accurately. This temporal standard is based on all kinds of data, but not on tetrapod skeletons or footprints, so is an independent standard against which tetrapod evolution can be calibrated. Taking advantage of this high-resolution chronological framework, and growing understanding of both the timing and extent of the CPE in this sector of eastern Pangaea[13,17,18], we here identify a significant Carnian shift in the composition of archosaur ichno-associations as recorded in the Southern Alps, which in turn provides the first well constrained date for the DDE in eastern Pangaea. Furthermore, we take advantage of new radiometric dating in western Pangaea[9,10] to integrate the oldest dinosaur-bearing formations of the South America in this model, providing a globally coherent scenario for the DDE. Finally, we highlight the synchronicity of the DDE and CPE and suggest a possible causal relationship between the two.

## Results and Discussion

**Background on the early history of dinosaurs.** The origin of Dinosauria is thought to have occurred in the mid- to high-latitudes of Gondwana[19,20] (but see ref. [21]). Although precise temporal calibration is to date unavailable, the Middle Triassic Manda beds of Tanzania yielded the remains of the possible oldest dinosaur, *Nyasasaurus parringtoni*[22], as well as the silesaurid *Asilisaurus*[23]. Therefore, even if *Nyasasaurus* is not a dinosaur, *Asilisaurus* is definitely a silesaurid, and Silesauridae is the immediate sister-group to Dinosauria, so minimally extending the origin of dinosaurs to the mid Anisian, about 245 Ma. These early dates have been confirmed by reports of non-dinosaurian dinosauromorph tridactyl footprints from the late Olenekian and Anisian of Poland[24], and other evidence from Italy, France, and Germany[25].

These Early and Middle Triassic dates for the origin of dinosaurs were unexpected and new[1]. Until 2010, the oldest undisputed members of Dinosauria were from the late Carnian of the lower Ischigualasto Formation of Argentina[26], whose age is radiometrically constrained between $231.4 \pm 0.3$ (Herr Toba bentonite) and $225.9 \pm 0.9$ Ma (Valle de la Luna Member)[26]. There, the presence of dinosaurs, such as *Panphagia*, *Eoraptor*, and *Herrerasaurus* in the basal horizons of the lower Ischigualasto Formation was used by Martínez and Alcober[27] to suggest that Dinosauria originated during the Ladinian or earlier and that they were already well diversified in the early Carnian. The *Hyperodapedon* Assemblage Zone (AZ) of the Santa Maria Formation of southern Brazil yielded some of the earliest dinosaurs (e.g., *Saturnalia* and *Staurikosaurus*[4]) recently redated at $233.23 \pm 0.73$ Ma[10]. Two further early dinosaur-bearing formations, the lower (and upper) Maleri Formation of India[28] and the Pebbly Arkose Formation of Zimbabwe[4] are less constrained in age, and are thought to be Carnian by biostratigraphic correlation within the *Hyperodapedon* AZ[4]. These skeletal records of early dinosaurs document a time when they were not numerically abundant, comprising typically <5% of individual specimens in their faunas, and when they were still of modest body size[2,3].

The DDE, indicated by dramatically increased relative faunal abundances worldwide and by body size increases in some forms, is documented by skeletal remains and footprints. The classic remains come from Europe. In the southern Germanic Basin, sauropodomorphs such as *Sellosaurus* and *Plateosaurus* dominate the Löwenstein and the Trossingen formations of mid-Norian age[29], both in terms of size and abundance. Theropods radiated

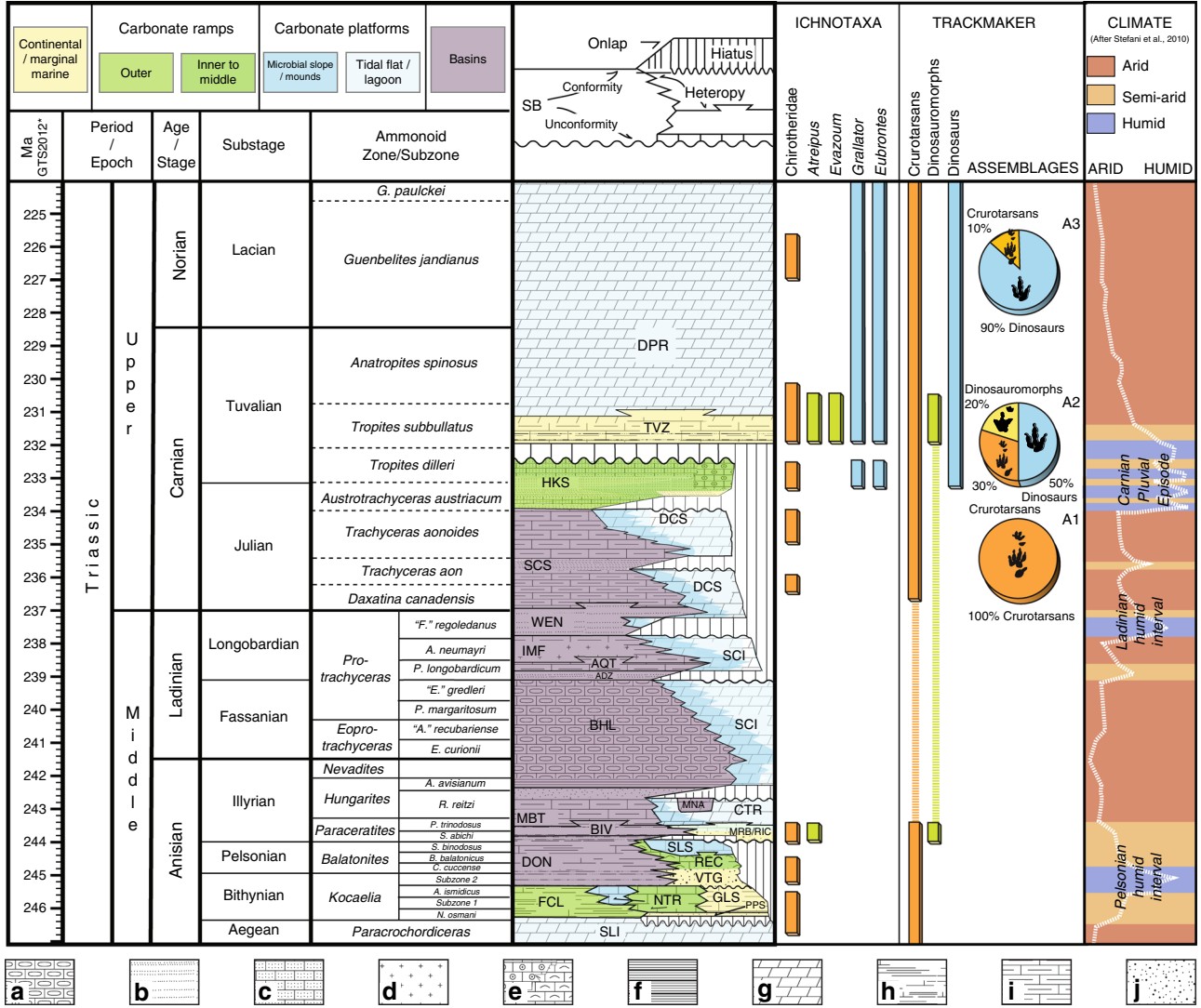

**Fig. 1** Stratigraphy in the Italian Dolomites, and dating the dinosaur diversification event. Footprint occurrences within the chrono-stratigraphic framework of the middle and upper Triassic of the Dolomites (Southern Alps, NE Italy), and the derived abundances of dinosaurian, non-dinosaurian dinosauromorph and crurotarsan trackmakers within the associations A1, A2, and A3 described in the text. Note the correlation between the change in composition of the associations and the climate shifts (climate trend and zones after ref. [11]). Lithostratigraphic abbreviations: ADZ: Zoppè Sandstone; AQT: Acquatona Formation; BHL: Livinallongo/Buchenstein Formation; BIV: Bivera Formation; CTR: Contrin Formation; DCS: Cassian Dolomite; DON: Dont Formation; DPR: Dolomia Principale; FCL: Coll'Alto dark Limestones; GLS: *Gracilis* Formation; HKS: Heiligkreuz Formation; IMF: Fernazza Formation and volcanites; MBT: Ambata Formation; MNA: Moena Formation; MRB/RIC: Richthofen Conglomerate and Morbiac dark Limestone; NTR: Monte Rite Formation; PPS: Piz da Peres Conglomerate; REC: Recoaro Limestone; SCI: Sciliar Formation; SCS: San Cassiano Formation; SLI: Lower Serla Dolomite; SLS Upper Serla Formation; TVZ: Travenanzes Formation; VTG: Voltago Conglomerate; WEN: Wengen Formation. Lithologies: **a** cherty limestone; **b** sandstone; **c** sandy limestone; **d** volcanics; **e** oolitic-bioclastic limestone; **f** black platy limestone or dolostone, black shale; **g** dolostone; **h** marlstone, claystone and shale; **i** marly limestone; **j** conglomerate. Ages from GTS timescale, modified after refs[12, 91, 92]. The silhouette images were created by the authors for use in this paper

too, with *Procompsognathus triassicus* and *Halticosaurus longotarsus* from the middle part of the Löwenstein Formation and *Liliensternus liliensterni* from the Trossingen Formation[29]. Of mid-late Norian age are also the Polish theropod skeletal findings recently discussed by Niedźwiedzki et al.[30]. The British Late Triassic *Thecodontosaurus*-bearing fissure faunas, once thought to be Carnian in age are now considered as Rhaetian[31]. The Ansbacher Sandstein (Stuttgart Formation, upper Schilfsandstein) and the Coburger Sandstein (Hassberge Formation) yielded some of the oldest definitive footprints assigned to dinosaurs[32] in this sector of Pangaea, which can be dated to the early to late Carnian (Julian-Tuvalian[33]). A single possible dinosaur footprint is known from the late Carnian of Monti Pisani in Central Italy[34]. Norian dinosaur footprints are then widespread in the Dolomia Principale/Hauptdolomit of the Alpine

region between Switzerland and Italy[25], and Poland-Slovakia (Tomanová Formation[35]). Large sauropodomorph tracks (*Eosauropus* and, possibly, *Evazoum*) have also been described recently from the Norian—early Rhaetian of Greenland[36].

In all, data from Laurasia show that dinosaurs were relatively rare in the Late Triassic of north and north-western Pangaea for some 10 myr after their Carnian occurrence in South America[2,5], and that they eventually became common from the Norian onwards. New data from the Dockum Group[37], and footprint findings in the Germanic basin[32], however, suggest that at least some dinosaur groups might have radiated synchronously across Pangaea[37].

The early dinosaur record of North America has been hard to date. The dinosaur-rich Petrified Forest Member of the Chinle

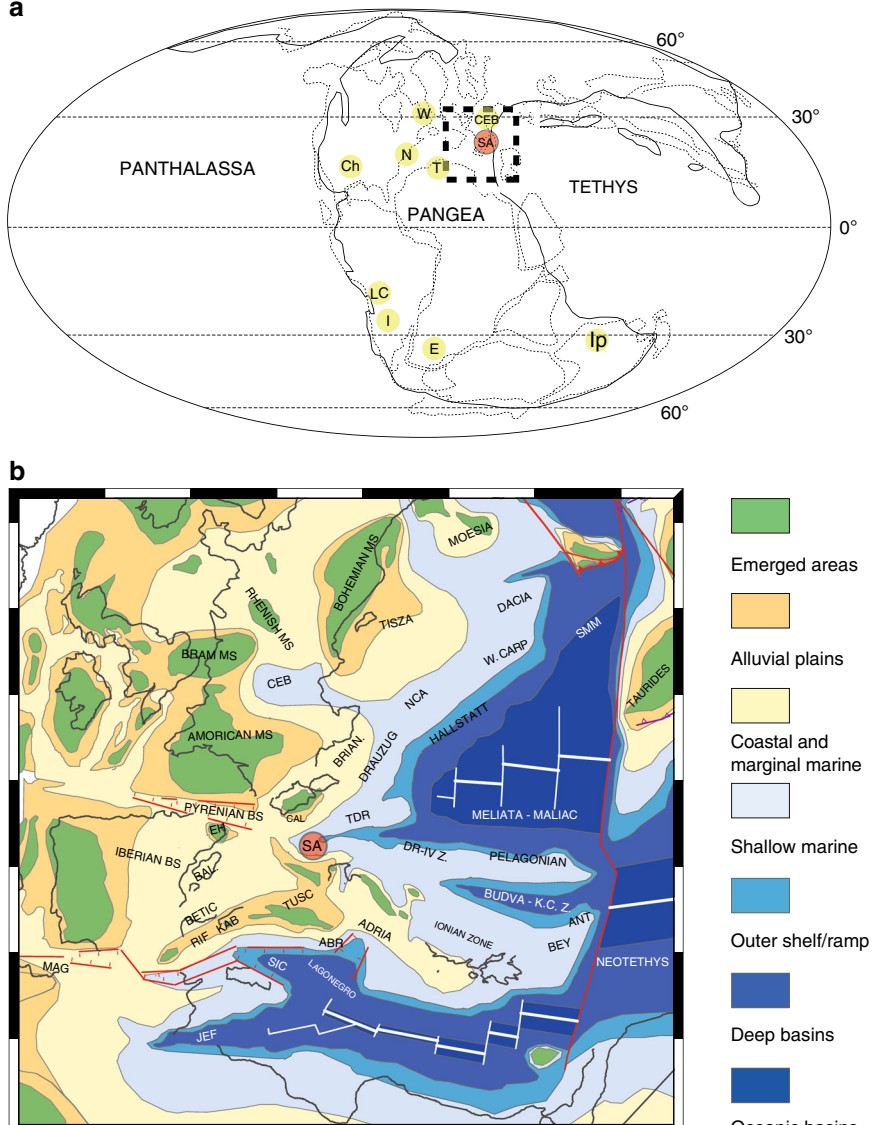

**Fig. 2** Early-late Carnian (Late Triassic) palaeogeographic reconstruction. **a** World map, showing palaeogeography in the late Triassic, and the main vertebrate-bearing units mentioned in the text. Abbreviations: CEB: Central European Basin; Ch: Chinle/Dockum Group; E: Molteno and Lower Elliot Formation (South Africa); I: Ischigualasto Formation; Ip: Ipswich Coal Measures (Australia); LC: Los Colorados Formation (Argentina); N: Newark basin; SA: Southern Alps; T: Timezgadiouine Formation (Morocco); Wolfville Formation (Nova Scotia, Canada). **b** Palaeogeographic map of western Europe showing land and sea, and main tectonic blocks. Abbreviations (BS: Basin; MS: Massif): ABR: Lazio-Abruzzi platform; ANT: Antalya; BAL: Balearic Islands; BEY: Beydaglari autochtonous; BRAB: Brabham Massif; BRIAN: Brianconnaise; CAL: Calabria; CARP Carpatian; DR-IV: Drina-Ivanica Zone; EH: Ebro High; JEF: Jeffara rift; K.C.Z.: Krasta-Cukali Zone; MAG: Maghrebian rift; NCA: Northern Calcareous Alps; RIFKAB: Rif-Kabylies nappe; SMM: Serbo-Macedonian Massif; TDR: Transdanubian Range; TUSC: Tuscany. The Fig. 2a map was modified from ref. [18] with permission from the Elsevier. All rights reserved. The Fig. 2b map was created by the authors using ArcGIS

Formation of southwestern North American, once thought to be of similar age to the Ischigualasto Formation, has now been re-dated to the late Norian-Rhaetian[5]. The early dinosaur record of North America is therefore mainly documented by the Newark Supergroup, the upper Chinle Formation, and the Dockum Group. The late Triassic Newark Supergroup of eastern North America has yielded a sparse tetrapod body fossil record and no dinosaur remains, although dinosaur footprints have been repeatedly signalled[38,39]. Olsen and Huber[40] reported the oldest dinosaur footprints from North America, dated to the late Carnian (Tuvalian), and associated with *Brachychirotherium*, and *Apatopus*. Cameron and Wood[41] described the co-occurrence of *Rhynchosauroides*, *Brachychirotherium*, *Atreipus* and dinosaur footprints (*Grallator*) in the upper portion of the Wolfville

Formation of Nova Scotia (Canada). Furthermore, Weems[42] documented some dinosaur tracks (*Eubrontes*) from the Balls Bluff Siltstone in the Culpeper Basin of Virginia, considered Carnian-Norian in age. *Camposaurus arizonensis*[43] is the oldest skeletal dinosaur evidence, from the lower part of the Chinle Formation, dated to ~220 Ma[44], while *Lepidus praecisio*[45] is the oldest dinosaur from the base of the Dockum Group, which is considered to be ?early Norian, although the age is only loosely constrained[45]. The Late Triassic Chinle Group yielded also abundant *Grallator* theropod tracks[39] and one possible *Eubrontes*[46]. All these records are most probably Norian in age[47].

In the southern hemisphere, Norian dinosaur remains are known relatively abundantly in the lower Elliot Formation of South Africa and the Los Colorados Formation of Argentina[4].

Dinosaur footprints (cf. *Eubrontes*) were reported from the Ipswich Coal Measures of Australia[48], which can be dated to the late Carnian[48] and that can be compared with similar findings from the Carnian-Norian Molteno Formation of South Africa[49]. The oldest skeletal remains from this sector of Gondwana are from the Lower Elliot Formation of South Africa which yielded basal dinosaurs, such as the sauropodomorphs *Euskelosaurus*, *Melanorosaurus*, *Blikanasaurus*, and *Antetonitrus*, and the ornithischian *Eocursor*[50]. The Los Colorados Formation is the source of the sauropodomorphs *Riojasaurus*, *Coloradisaurus*, and *Lessemsaurus*, and the theropods *Zupaysaurus* and *Powellvenator*. The coeval, or slightly younger, Quebrada del Barro Formation has produced a similar fauna, comprising a theropod, basal sauropodomorphs, and other reptiles, while the Laguna Colorada Formation (El Tranquilo Group) has yielded the sauropodomorph *Mussaurus* and a heterodontosaurid ornithischian[4]. The ages of the Lower Elliot, Quebrada del Barro, and Laguna Colorada formations are however poorly constrained to the Norian[4]. In the Lower Elliott Formation (Norian-Rhaetian), Ellenberger[51] reported also a variety of footprint morphotypes, some of which can be attributed to dinosaurs[52]. Of probable late Carnian age are the dinosaur footprints discovered in the Timezgadiouine Formation of Morocco and assigned to *Eubrontes* and *Grallator*[53].

**Chronological constraints in the Dolomites region.** The stratigraphy of the Southern Alps and of the Dolomites in particular encompasses the whole Triassic system, providing an extraordinary record of the environments and ecosystems constrained by an excellent bio-chrono-stratigraphic framework (Fig. 1), particularly for the early Anisian to late Carnian interval, which has made this area a reference worldwide for the Triassic timescale. During this time, the Southern Alps (Fig. 2) were located at tropical latitudes in the western Tethys region (about 15–18° North[54]), which was characterized by a complex palaeogeography, with narrow ocean branches separated by carbonate platforms or emerged lands. Overall, the Southern Alps faced an ocean towards the present-day east and were characterised by the presence of an emerged land (Adriatic foreland) towards the present-day south[16]. The Southern Alps area was a wide shallow sea during the late Permian and Early Triassic, but started to differentiate at the beginning of the early-middle Anisian. Later, a sudden increase in subsidence combined with a strong sea level rise allowed a general deepening, associated with the formation of high-relief carbonate buildups and a general retreat of the siliciclastic shoreline. Subsidence rates reached a climax during the late Anisian, and at that time the palaeogeography of the Dolomites and Lombardy featured numerous small isolated carbonate platforms (Sciliar and Esino formations) surrounded by a deep basin (Buchenstein Formation).

From late Ladinian to early Carnian, the subsidence rate decreased, resulting in the progradation of the southern shoreline and a general shallowing of the basins. At that time, the palaeogeography was characterized by large emerged areas to the south-south-west, bounded by attached carbonate platforms and small isolated platforms in the north-eastern Southern Alps[55]. This regressive trend culminated in the late Carnian, with a strong north-eastward shift of the coastline and complete flattening of the palaeotopography, matched by a climatically driven increase in siliciclastic input[56], restoring a relative homogeneity in sedimentary palaeoenvironments[57], as documented by the Heiligkreuz and Travenanzes formations. A new transgression in the latest Carnian allowed the deposition of the thick peritidal succession of the Dolomia Principale, which records a huge regional platform that extended for hundreds of kilometres from north to south and east to west[58]. Widespread carbonate platforms then characterised the Southern Alps for several million years, from the late Carnian to (at least) the late Norian.

Primary and secondary palaeoclimate indicators[56,59] suggest a generally arid climate with elevated temperatures, interrupted by short "humid shifts" (Fig. 1). The most important humid episode occurred during the early Carnian. This episode, marked at the base by a sharp negative carbon-isotope excursion of about 4‰[17,18], is documented by multi-proxy evidence of increased rainfall as a sudden increase of coarse and immature siliciclastics into the basin, humid paleosols, hygrophytic pollen assemblages, and massive resin production[60,61]. It is also associated with the abrupt demise of high-relief microbial carbonate platforms[59]. All these factors have been interpreted as regional evidence for the CPE[17,18,56].

It should be noted that the CPE is characterised in the Italian Dolomites by at least four humid pulses before the return to arid conditions[59,61] and has great biostratigraphic control[18,56,60,61], being constrained between the *Aonoides/Austriacum* boundary interval (about Julian) and the base of the *Subbullatus* Zone (Tuvalian), an interval of about 1.6/1.7 myr around 234–232 Ma (Fig. 1).

The CPE, documented in the Dolomites by the Heiligkreuz Formation, can be matched in other palaeogeographic settings around the western border of the Tethys Gulf with the Lunz Formation and the Raibler Schichten in the Northern Calcareous Alps[61,62], the Veszprem Marls and the Sandhoregy Formation in the Trans-Danubian Range[18] and the Stuttgart Formation and equivalents in the Germanic basin[33]. The marine Southern Alpine record of the CPE can be compared with coeval terrestrial British facies, where several thin sandstone units sit within the mainly mudstone-dominated Mercia Mudstone Group of the West Midlands, Somerset, and Dorset. Borehole records of the latter show at least five short-lived carbon isotope excursions spanning an estimated 1.09 myr[63], all of which provide evidence of a good match with the records from Italy. Although some regions of Pangaea are still understudied, the CPE was most probably a global event[15] that corresponds to one of the most severe biotic crises in the history of life[64].

**A Carnian dinosaur shift in tetrapod ichno-associations.** In the Italian Southern Alps, the record of tetrapod ichno-associations is more or less continuous from the late Carboniferous to early Jurassic. Only during the latest Permian, latest Anisian and Ladinian is there no evidence of tetrapod traces from the Southern Alps, associated with the general retreat of the shoreline and/or the presence of isolated platforms. In the Julian to Norian interval, attached platforms and marginal marine environments became widespread in the region, associated with the common peritidal environment. Here, various groups of tetrapods became common, as evidenced by the numerous ichnosites discovered in the last decades. Three main ichnoassemblages can be distinguished: a Crurotarsi-dominated assemblage in the Julian, early Carnian (A1), a mixed assemblage in the Tuvalian, late Carnian (A2) and a dinosaur-dominated assemblage in the late Tuvalian–Norian (A3) (Fig. 1).

Assemblage 1: Crurotarsi-dominated, Julian. Five ichnosites of Julian age record the presence of tetrapods in the region. The Val Sabbia Sandstone Formation yielded well-preserved trackways assigned to the ichnogenus *Brachychirotherium*, known to have been produced by quadrupedal crurotarsan archosaurs[65]. Crurotarsan tracks (Chirotheriidae) were reported[66] in the uppermost Cassian Dolomite (Aonoides Zone, Julian), as in the Busa dei Cavai (Mondeval), Nuvolau and Settsass sites. In all, these Julian

sites provide evidence for crurotarsan dominance (80–100%) of early Carnian tetrapod associations in the Southern Alps region.

Assemblage 2: Mixed Crurotarsi—Dinosauria, early Tuvalian. Six ichnosites of early Tuvalian age record mixed archosaur ichno-associations and the oldest occurrence of dinosaurs in the region. Small tridactyl footprints assigned to the ichnogenus *Grallator* and attributed to theropod dinosaurs are dominant in the Travenanzes Formation of the San Gottardo site[67] and co-occur with the tridactyl *Atreipus*-like footprints, attributed to dinosauriforms[67], and the tetradactyl *Evazoum*, attributed either to sauropodomorphs[68] or crurotarsans[69]. A single-pentadactyl track assigned to *Brachychirotherium*[70] testifies to the presence of crurotarsan archosaurs. Crurotarsan archosaurs (e.g., *Brachychirotherium*) also occur in the Heiligkreuz/Travenanzes formations of the Mostizzolo ichnosite[71], together with a large *Eubrontes* track, attributed to a dinosaurian trackmaker[70]. Large dinosaur footprints (foot length = 26–27 cm), attributed to *Eubrontes*, were also described by Bernardi et al.[70] from the Monte Roen site of similar age. These specimens indicate the presence of large dinosaurs, ca. 5 m long[70]. Footprints referred to prosauropods have been reported[66] in situ in the Heiligkreuz Formation of Lastoi di Formin and in erratic blocks in the locality Vare di Giau (Giau Pass); a single tridactyl footprints attributed to a small theropod has been found in the Sasso della Croce (Heiligkreuz Formation). Unnamed archosaurian footprints showing affinities with the ichnogenus *Brachychirotherium*, associated with nesting structures, are also known from the Monticello Member (Dolomia Principale) of Tuvalian age in the Dogna Valley site[72]. Finally, an enigmatic quadrupedal trackway from the Ciol de la Fratta site (Carnian Pre-Alps), of late Carnian age, has been referred to a large crurotarsan trackmaker[73]. A similar mixed association has been described also from the Carnian of Lerici (La Spezia)[68], but the lack of precise temporal constraint prevents any further discussion of these data, and they are therefore omitted from calculations. In all, early Tuvalian sites provide evidence for the oldest dinosaurs in western Pangaea and for a mixed faunal composition, with 40% of specimens being dinosaur tracks, 10% non-dinosaurian dinosauromorph tracks, and 50% crurotarsan tracks.

Assemblage 3: Dinosaur-dominated, late Tuvalian-Norian. More than ten ichnosites of late Tuvalian to Norian age, all in the Dolomia Principale, are evidence for the abundant presence of dinosaurs in the Southern Alps. Numerous dinosaur footprints were described[74] from the Mt. Pelmetto ichnosite, which also yielded a single *Brachychirotherium*-like footprint[75], and another in the same rock slide[66]. The trampled horizon can be constrained to the lowermost part of the Dolomia Principale directly overlying the Travenanzes Formation, and can be dated to the late Tuvalian[57]. A *Eubrontes-Grallator* association was also described[76] from the Tre Cime site, and dinosaur-only associations were described also in the Friuli and Carnic Prealps[77]. All these can be dated to the late Carnian-Norian by means of stratigraphic position. An association of *Evazoum*, *Eubrontes* and *Grallator* was also reported[78] from the Pasubio Massif, whose Norian (Alaunian) age is established by conodont biostratigraphy. A mixed chirotheroid—dinosaur footprint association was also reported from the Val Pegolera[66] outcrop and a dinosaur-dominated *Atreipus-Grallator* association was described from the Moiazza ichnosite[66]. All late Triassic ichnosites of late Tuvalian to Norian age provide evidence for dinosaur dominance, with >90% of tracks being assignable to dinosaurian trackmakers.

**Dinosur diversification and the Carnian Pluvial Episode.** The early evolution of dinosaurs has recently attracted much interest, with new discoveries[79], new phylogenies[20,21] and new theoretical models and computational tools[1] radically enhancing our understanding of the tempo and mode of origination, early diversification, and dispersal of this group. As discussed above, however, with a few notable exceptions[5,6,9], the lack of independent age dating for most specimens has hindered any attempt to reconstruct the precise sequence of events, blurring our understanding of the earliest phases of dinosaur evolution.

The occurrence of several precisely dated ichnoassemblages in the late Triassic of the Southern Alps allows us to constrain the timing of the DDE in this region of Pangaea. In a relatively brief interval of about 3-4 myr (around early/late Carnian), dinosaurs shifted from near or complete absence (0% in the Cassian Dolomite/Val Sabbia Sandstone association of Julian age) to notable presence (ca. 40% of the Heiligkreuz-Travenanzes formations association of early Tuvalian age), to ecological dominance (>90% of the Dolomia Principale association of late Tuvalian and Norian age). Although new discoveries might slightly modify these percentages, it is improbable that the differences between the three assemblages could be the result of sampling biases, because no single dinosaur footprint has ever been found below the CPE in the Southern Alps and very few crurotarsan footprints have been found after the CPE, despite intensive sampling in the last 40 years. Further, this percentage shift in the relative abundances of dinosaurs estimated from tracks and trackways confirms figures noted by earlier authors[2,3] based on counts of skeletons, and showing a shift in relative dinosaurian abundance from 5–10% to 70–90% through the DDE.

The explosive increase in dinosaurian abundance in terrestrial ecosystems, which had been dated variously to the Carnian and Norian in other parts of the world, is therefore constrained in the Southern Alpine region to the early-Late Carnian (early Tuvalian), an order-of-magnitude improvement of dating precision over most earlier work.

The U-Pb dating of the Argentinian Chañares Formation has been used[6] to suggest that the shift from assemblages containing only dinosaur precursors to those with early dinosaurs occurred, in the high latitudes of Gondwana, between the early and late Carnian and took <5 myr. Similarly, recent U-Pb dating of the Santa Maria Formation and the Caturrita Formation in Brazil[10] constrained the first dinosaur diversification in the region between 233 and 225 Ma. Our analysis of the Italian Dolomites supports this timing and provides a high-resolution chronological framework for this event. It also suggests that the first major dinosaur diversification might have been a synchronous event across all Pangaea. The oldest widespread dinosaur evidence is provided by footprints recorded in the Heiligkreuz Formation of the Southern Alps, in the Ansbacher Sandstein of the Germanic Basin[32] and in the Los Rastros Formation in Argentina[80], which can all be dated to the late early Carnian and is soon followed by skeletal evidence in western Pangaea[6,10] (Fig. 3). Both in Gondwana (Ischigualasto, Caturrita, Los Colorados, Lower Elliot formations) and Laurasia (Dolomia Principale, Löwenstein, Trossingen formations), dinosaurs then dominated through the Norian[81] (Fig. 3).

The CPE[14,15,17] marks an important phase of climate destabilisation. Abrupt environmental changes, such as warming, ocean acidification, mega-monsoonal conditions, and a generalised increase in rainfall are observed in the geological record worldwide during this time[15,18,82] and these phenomena were synchronous with a carbon-cycle perturbation[17,18] that could be linked to the Wrangellia Large Igneous Province volcanism[17]. The CPE is characterised by elevated extinction rates in several marine groups such as crinoids, scallops, corals, ammonoids, and conodonts, and an abrupt interruption in organic carbonate production in shelf settings (reviewed in ref. [15]); it is considered

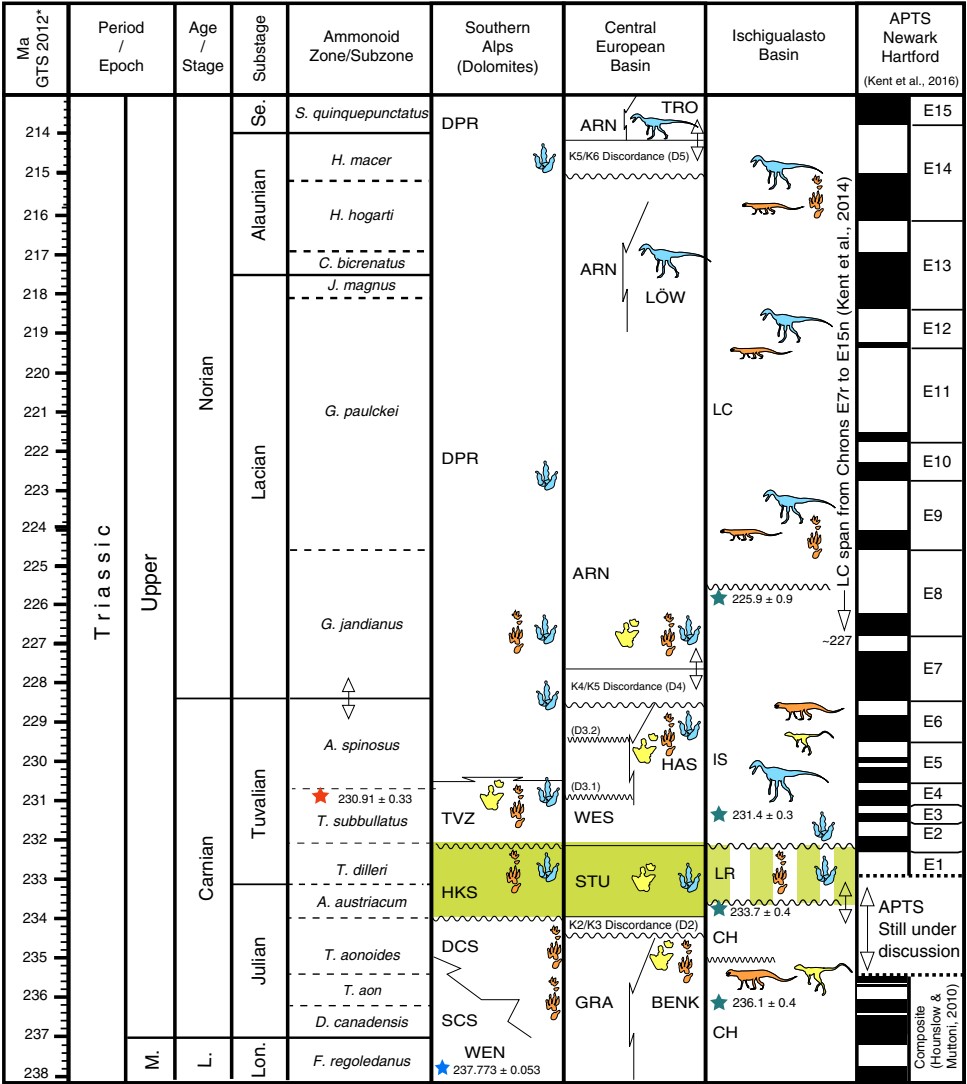

**Fig. 3** Correlation between the earliest dinosaur occurrences across Pangaea. Note the synchronicity of the first dinosaur diversification event during and after the CPE (light green boxes). Lithostratigraphic abbreviations. Dolomites: DCS: Cassian Dolomite; DPR: Dolomia Principale; HKS: Heiligkreuz Formation; SCS: San Cassiano Formation; TVZ: Travenanzes Formation; WEN: Wengen Formation. CEB (Central European Basin): ARN: Arnstadt Formation; BENK: Benk Formation; GRA: Grabfeld Formation; LÖV: Löwenstein Formation; STU: Stuttgart Formation; TRO: Trossingen Formation; WES: Weser Formation. Ischigualasto Basin: CH: Chañares Formation; IS: Ischigualasto Formation; LC: Los Colorados Formation; LS: Los Rastros Formation. Lithostratigraphy of CEB from Stratigraphische Tabellen von Deutschland 2016 (http://www.stratigraphie.de/std/Bilder/5_2.pdf). Red Star: Aglianico-Tuff, Lagonegro Basin[13]; Blue Star: Rio Nigra Tuff, Dolomites[93]; Green Star: Radiometric ages from Ischigualasto Basin (from Ezcurra et al.[9] and those published in ref. [6]). Archosaur silhouettes: skeletal data; Footprint silhouettes: ichnological data; Light blue: dinosaurs; Yellow: non-dinosaurian dinosauromorphs; Orange: crurotarsans. Fossil occurrences from refs. [4, 6, 9, 25, 29, 32, 65–73, 76–78, 80]. Ages as in Fig. 1. The silhouette images were created by the authors for use in this paper

among the most severe biotic crises in the history of life[64]. On the other hand, this event coincides with the appearance of the first abundant pelagic calcifiers and of scleractinian corals[83,84], the radiation of modern conifers[85] and the diversification of archosaurs which replaced the then-abundant herbivorous rhynchosaurs and dicynodonts[3], possibly reflecting the diversification of conifers and the decline of *Dicrodium* seedferns.

Data presented here suggest that the first dinosaur dispersal in eastern Pangaea and the DDE are synchronous with the CPE and that dinosaurs became dominant only after this perturbation (Figs. 1 and 3). In the Southern Alps, in fact, dinosaurs are absent in the Julian formations, which are older than the CPE (Cassian Dolomite and Val Sabbia Sandstone; ca. 236 Ma), and appear in the overlying Heiligkreuz-Travenanzes formations, which are coeval or slightly younger than the Episode (ca. 234 Ma) to

become dominant just a few million years later, at the base of the Dolomia Principale (ca. 230 Ma).

This pattern is also matched by the first occurrence of dinosaurs in the Germanic Basin[32], within the Ansbacher Sandstein (Stuttgart Formation, upper Schilfsandstein), which correlate with the Heiligkreuz Formation and represent the regional expression of the CPE. The same pattern can also be recognized in South America, where the first occurrence of dinosaurs is in the Los Rastros Formation[80]. This unit, which records a sharp shift from fluvial to lacustrine, and then back to fluvial conditions, unconformably follows the Chañares Formation, recently[9] assigned to the Carnian through radioisotopic dating of detrital zircons (ages 233.7 and 236.1 Ma). The Los Rastros Formation is overlaid by the Ischigualasto Formation, which is constrained by a $^{40}Ar/^{39}Ar$ date of 231.4 ± 0.3 Ma from a

tuff near the base of the unit. This chronostratigraphic framework allows correlation of the continental sections of South America and the Tethyan sections (Fig. 3). Moreover, the most recent radioisotope dating[6,9] and magnetostratigraphic correlation with the Newark astrochronological polarity timescale[81] strongly support the synchronicity between the humid event as recorded in Europe (Fig. 3) and the biotic turnover recorded in the various basins. We hasten to point out that the signature of the CPE in Southern Pangea has yet to be verified through detailed stratigraphic studies. However, we note that the shift to more humid environments is documented in the Los Rastros Formation both by sedimentological and palynological evidence[86,87], that this formation records the earliest evidence of dinosaurs[80], and that it is overlain by the Ischigualasto Formation where dinosaurs became abundant (Fig. 3), mirroring the pattern and showing temporal synchronicity with the eastern Pangaea record and in particular with the Southern Alpine Heiligkreuz and Travenanzes formations, where dinosaurs appeared and started to become dominant, providing evidence for possible comparable macroevolutionary dynamics throughout the whole of Pangaea. Notably, the first dinosaur diversification in southern Brazil, which occurred in the late Carnian[10], was recently suggested to be linked with the climatic oscillations of the CPE, although no conclusive supporting geological evidence is to date available[10]. Our model supports these intuitions and provides a new framework for interpreting these and other early dinosaur-bearing faunas.

To date, uncertainties in dating of the best known early dinosaur association obscured this pattern (but see refs. [10,88]) and prevented a test for coincidence between the CPE and the DDE, although geologists repeatedly hypothesised this[13]. Unfortunately, the chronostratigraphic precision available in the Southern Alps and partially in South America is currently unavailable in other parts of the world, so preventing a global verification of this hypothesis at the moment.

More studies are needed to demonstrate a causal link between the CPE and the DDE, but we note that the link is plausible in that both environmental factors (e.g., a more humid tropical belt, and more emergent land created by the infilling of the basins) and biological factors (high turnover in ecosystems, vegetation change) are possible drivers of a rapid dispersal and diversification of the dinosaurs.

Finally, we note that the most recent palaeogeographic models suggest that dinosaurs diversified at middle to high palaeolatitudes[6,19,20] and that they eventually became dominant at tropical palaeolatitudes much later, possibly in the Norian. However, the well dated evidence presented here indicates that dinosaurs were present in northern Pangaea at least from the late Carnian, and that soon after their arrival in the region they became dominant in their ecosystems. The coexistence of dinosaurs, dinosauriforms and crurotarsan archosaurs, therefore, was also more prolonged than thought, and began at least in the middle Carnian. This provides support for the view that crurotarsan-dominated faunas were substituted by a gradual process of ecological replacement[1,2,23,26] that might have been accelerated by the ecological reshuffling ignited by the Carnian Pluvial Episode, which triggered the extinction of key herbivores, including rhynchosaurs and most dicynodonts[3,8], and the first diversification of dinosaurs.

## Methods

**Identification criteria**. We reviewed all published tetrapod tracks described in the last decades from the Southern Alps and re-assessed their stratigraphic positions and ages, based on the most recent biostratigraphic schemes. Ichnogenera/morphotypes were attributed to three major groups (i.e., crurotarsans, non-dinosaurian dinosauromorphs, and dinosaurs) on the basis of personal study and published papers.

The attribution of Mesozoic tridactyl prints to Dinosauria is customary[52], although possibly incorrect. This morphotype cannot in fact be unambiguously assigned to dinosaurs as at least some non-dinosaurian dinosauromorphs possessed a functionally tridactyl pes[89]. Dinosaur tracks are however recognised on the basis of several synapomorphies: (i) dominance of the digit II–IV group, (ii) mesaxonic pattern of foot structure, (iii) digit I reduced and shifted backwards (and thus often not preserved in tracks), (iv) bunched metatarsus, and (v) tendency towards digitigrady[2,90]. These characters are present in all dinosaur footprints cited herein. Furthermore, although we do not discuss any specific attribution within Dinosauria, theropod-like footprints can be recognised on the basis of the following characters: (i) asymmetry of the track, with angle between digit III and II lower than between III and IV, (ii) digit III longer than IV > II, (iii) sharp claw traces on all digits, (iv) tip of digit II turned inwards, and (v) bipedalism[70]. Presence of these characters in the specimens studied here is therefore at least supporting evidence for a dinosaurian producer.

The abundance of the various trackmakers in the ecosystems has been calculated as the percentage of specimens attributed to each trackmaker for each site. We considered a "specimen" as each single evidence of the presence of a tetrapod, whether a track (=single print) or a trackway (=multiple prints) assignable to a single ichnotaxon.

**Data availability**. The authors declare that all data generated or analysed during this study are included in this published article.

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

## Acknowledgements

Matthias Franz, (Göttingen University) Edgar Nitsch (LGRB Freiburg) and Manfred Menning (GFZ Postdam) are thanked for useful information of the Germanic Basin; Nereo Preto, Manuel Rigo (Padova University) and Jacopo Dal Corso (Leeds University) are thanked for fruitful discussions on the CPE; and Guido Roghi (CNR-IRPI) for useful comments on pollen association. Hendrik Klein (SPM Neumarkt) and Marco Avanzini (MUSE Trento) provided useful comments on the tetrapod ichno-associations. P.G. thanks Marcello Caggiati (Ferrara University) for the palaeogeographic map of the upper Carnian. M.B. thanks La Sportiva for supporting field activities in the Dolomites. Funded in part by the NERC BETR grant NE/P013724/1 to M.J.B. and by PRIN 2010-2011 (Pr. No 20107ESMX9_004) to P.G.

## Author contributions

M.B. and P.G. designed the study. M.B., F.M.P., and M.J.B. developed the palaeontological parts of the study, while P.G. and P.M. contributed in the more geological sections. All authors interpreted the results. M.B. and P.G. led the writing of the paper and all other co-authors contributed to the final version.

## Additional information

**Competing interests:** The authors declare no competing interests.

