## [Peer Review File · Nature Communications]

Reviewers' comments:

Reviewer #1 (Remarks to the Author):

This contribution includes an up-dated and thorough discussion about the stratigraphy of the Italian southern Alps, focus on its well-known and abundant footprint record, and also the chronostratigraphic framework during the Middle-Late Triassic in the region is considered. The final discussion based on the shift of the identified ichnoassemblages across the Carnian-Norian and its postulated relation to the Carnian Pluvial Episode is quite compelling. Up to this point, I think the Ms is significant to a broad spectrum of paleontologists and geologists, not just Triassic dinosaur researchers.

Nevertheless, there are some significant things that need to be resolved before acceptance. The major issue is in the general introduction to the Ms (lines 36-86) and particularly, the text under "The early history of dinosaurs" section. The text contains a mix discussion of dinosaur faunas around the world thus leading to what can best be described as circular reasoning. Therefore, in several paragraphs, they discuss chronostratigraphic ages (date) of dinosaur faunas based on biostratigraphy, such as the Manda beds of Tanzania (lines 86-92). These beds are, in fact, devoid of both published radioisotopic dates and other independent criteria for assessing their age, and were "dated" based on biostratigraphic correlations with the Cynognathus AZ of the main Karoo Basin, which also lacks any published radioisotopic dates. This "practice" continues with other Gondwanan dinosaur-bearing levels, as the Maleri Fm. of India and the Pedbbly Arkose Fm. of Zimbabwe (in lines 105-110).

Accordingly, the whole section is convoluted and do not help to conduct finally the reader to the main goal of the paper. I consider that the general introduction to the Ms should be rethink and focused on those dinosaur faunas with skeletal remains that have been confidently dated (absolute ages) such as those from the Ischigualasto Basin in Argentina and the Chinle Fm. of USA, and thus compare them with their southern Alps discussed data. In fact, this is what the authors do at the end of the contribution, in the final figure (Figure 3)!

Reviewer #2 (Remarks to the Author):

G'day,

It has been a pleasure to review the drafted manuscript Dating Dinosaur Diversification for Nature Communications. The scientific merit of the manuscript is potentially high; however, in its present form, this manuscript requires major revisions or additions to its present form. Fundamentally, this manuscript tries to address: 1) possible first occurrence age (FOA) of dinosaurian lineages; 2) if it co-occurred with other trigger events (abiotic or biotic); and 3) if this was a singular or a staggered event. As a reviewer, I agree that this FOA needs reassessment; yet, I feel that the author(s) need to develop a more robust framework within this manuscript to properly interpret these three points based on the current data.

General Assessment:

- Should the manuscript be published: Yes, WITH REVISIONS
- Is the work original: Yes
- Scientific method present: Yes, but requires development
- Conclusion: Manuscript requires development

Specific Comments

- Manuscript requires grammatical and typographical development
- Geochronological data is thin, poorly compared. Author could better the descriptive text on how they

assessed each radiometric ages for units, ichnites or assemblages

- Radiometric age assessments and temporal placements of stratigraphic units or ichnite assemblages is based on very limited results causing flimsy correlations
- Very broad implications with thin data within text to support it, author(s) could better the correlations/comparisons by including more lithostratigraphic and basinal context
- The narrative is currently illustrating narrowly focused results with broad sweeping assumptions on limited to fragmentary global correlations, this needs development
- Author(s) need to develop a better argument for commonly occurring biases in the fossil record that may affect their interpretations
- Author(s) commonly suggest "our data" (such as Line 323), but linkages are loose and the author needs to develop this. The author(s) need to more clearly identify what they are contributing in terms of scientific knowledge or results and the means by which they achieved them

- With development of the narrative and a strengthening of data to support the proposed results, this paper would make a great contribution to our understanding of early dinosaurian evolution

Reviewer #3 (Remarks to the Author):

Dear Editor Nature Communications,

I congratulate the authors for such a well-written and clear MS with well-exposed hypotheses and ideas. The contribution by Bernardi et al. is really important and of deep importance and would result of broad interest on the paleontological community. The MS includes new evidence regarding the origin and early evolution of dinosaurs, based on novel evidence.

I think that the MS should be published after minor review (I attached a PDF including some citations that, I think, authors should include in the final version of their contribution.

The statistical analyses are well-founded and are sound.

All the best,

Federico Agnolin

[Editor Note: Due to journal policy, we are unable to publish the Reviewer #3 annotated manuscript file as part of this Peer Review File.]

Reviewer #1 (Remarks to the Author):

This contribution includes an up-dated and thorough discussion about the stratigraphy of the Italian southern Alps, focus on its well-known and abundant footprint record, and also the chronostratigraphic framework during the Middle-Late Triassic in the region is considered. The final discussion based on the shift of the identified ichnoassemblages across the Carnian-Norian and its postulated relation to the Carnian Pluvial Episode is quite compelling. Up to this point, I think the Ms is significant to a broad spectrum of paleontologists and geologists, not just Triassic dinosaur researchers.

Many thanks. We highlight this is the main goal of our paper and we appreciate it has been recognized as an original and convincing piece of work.

Nevertheless, there are some significant things that need to be resolved before acceptance. The major issue is in the general introduction to the Ms (lines 36-86) and particularly, the text under "The early history of dinosaurs" section. The text contains a mix discussion of dinosaur faunas around the world thus leading to what can best be described as circular reasoning. Therefore, in several paragraphs, they discuss chronostratigraphic ages (date) of dinosaur faunas based on biostratigraphy, such as the Manda beds of Tanzania (lines 86-92). These beds are, in fact, devoid of both published radioisotopic dates and other independent criteria for assessing their age, and were "dated" based on biostratigraphic correlations with the Cynognathus AZ of the main Karoo Basin, which also lacks any published radioisotopic dates. This "practice" continues with other Gondwanan dinosaur-bearing levels, as the Maleri Fm. of India and the Pebbly Arkose Fm. of Zimbabwe (in lines 105-110).

This is a sort of introductory section. We felt it was necessary to set the context and to provide an updated state of the art to the non-expert. The other reviewers were content with this introductory portion, in which – it should be noted – we reviewed existing literature and existing understands. Indeed, some of the issues of circularity mentioned by the reviewer are the issues we argue against, and this justifies the introduction of the new evidence from the Italian Dolomites, where we argue we can cut through the Gordian Knot of circularity because the Italian sections are dated 100% independently of the tetrapod data (more in the next comment-reply below).

Accordingly, the whole section is convoluted and do not help to conduct finally the reader to the main goal of the paper. I consider that the general introduction to the Ms should be rethink and focused on those dinosaur faunas with skeletal remains that have been confidently dated (absolute ages) such as those from the Ischigualasto Basin in Argentina and the Chinle Fm. of USA, and thus compare them with their southern Alps discussed data. In fact, this is what the authors do at the end of the contribution, in the final figure (Figure 3)!

Quite so – agreed. We added a few sentences into the Introduction to make this explicit: ‘This highlights the need to avoid circularity in dating the key events in the origin of dinosaurs by not using tetrapod faunas to provide the dating. We believe we have the solution here, which is to use rock sections that are dated independently of the tetrapod fossils, by means of magnetostratigraphy, radioisotopic dates, and correlations to marine successions dated by ammonites, conodonts and other fossils.’

Reviewer #2 (Remarks to the Author):

G'day,

It has been a pleasure to review the drafted manuscript Dating Dinosaur Diversification for

Nature Communications. The scientific merit of the manuscript is potentially high; however, in its present form, this manuscript requires major revisions or additions to its present form. Fundamentally, this manuscript tries to address: 1) possible first occurrence age (FOA) of dinosaurian lineages; 2) if it co-occurred with other trigger events (abiotic or biotic); and 3) if this was a singular or a staggered event. As a reviewer, I agree that this FOA needs reassessment; yet, I feel that the author(s) need to develop a more robust framework within this manuscript to properly interpret these three points based on the current data.

Thank you for the appreciation; we shall make revisions as suggested.

General Assessment:

- Should the manuscript be published: Yes, WITH REVISIONS
- Is the work original: Yes
- Scientific method present: Yes, but requires development
- Conclusion: Manuscript requires development

We shall make all suggested revisions.

Specific Comments

- Manuscript requires grammatical and typographical development

Agreed – we have further revised the whole MS with close attention to clarity, eliminating repetition and waffle, and improving the English expression.

- Geochronological data is thin, poorly compared. Author could better the descriptive text on how they assessed each radiometric ages for units, ichnites or assemblages

Agreed. We have added clear statements about the sources of data, and thorough cross-referencing on radioisotopic dates and magnetostratigraphic arguments. We also added one more reference (n. 92) to provide another recent evidence of the progress in the field. We highlight we do not present any original data here, but derive our information from the rich recent literature on these topics, and we provide clear referencing for each statement.

- Radiometric age assessments and temporal placements of stratigraphic units or ichnite assemblages is based on very limited results causing flimsy correlations

As noted, we do not provide any new radioisotopic dates and magnetostratigraphic arguments, but we interpret the detailed and specialist literature, which is thoroughly referenced throughout the MS (within the limits of *Nature communications* rules). We include all the current information, and we would dispute the reviewer's use of the term 'flimsy' – we reference literature running to thousands of pages, and coming from leading laboratories in the United States, Italy, the UK, and elsewhere.

- Very broad implications with thin data within text to support it, author(s) could better the correlations/comparisons by including more lithostratigraphic and basinal context

We provide great details, in the paper and in referencing dozens of recent papers on the Dolomites basins, providing the most thorough detail on biostratigraphy, magnetostratigraphy, and independent dating. The co-authors have done much of this work, and so they are intimately acquainted with it. We profit also from the excellent new work by leading geologists on new magneto/radioisotopic dating in South America, North America, Africa, Europe, and elsewhere.

- The narrative is currently illustrating narrowly focused results with broad sweeping assumptions on limited to fragmentary global correlations, this needs development

We have clarified the sources of data for the key Dolomites sections, and each point is fully developed in the referred literature, much of it led by co-authors of the current paper. The reviewer does not provide examples of where we have failed to include any current work or to demonstrate additional clarity on stratigraphy.

- Author(s) need to develop a better argument for commonly occurring biases in the fossil record that may affect their interpretations

We have taken thorough account of biases and gaps in knowledge. Our work is based on the most completely documented sections in the world for this time interval, namely in the Dolomites and in the Ischigualasto Basin in Argentina. These are the best, but even so, of course, suffer many gaps. We stressed this even more in the revised version of the paper, and refer to all relevant literature on these points.

- Author(s) commonly suggest “our data” (such as Line 323), but linkages are loose and the author needs to develop this. The author(s) need to more clearly identify what they are contributing in terms of scientific knowledge or results and the means by which they achieved them

We clarified this point by addition of the phrase ‘Our data from the new model of correlations and dating in the Italian Dolomites...’ This criticism is similar to earlier criticisms. We have combed our text carefully, checking for any lack of clarity in citation. There is not space in a *Nature Communications* article to repeat all the thousands of pages of recent work done by us and our collaborators in the Italian Dolomites, so we reference the key 20 recent papers in which results are presented in more detail, with descriptions of field outcrops and geochemical, palaeontological, and geophysical analyses already completed – the basis of our work summarised here.

- With development of the narrative and a strengthening of data to support the proposed results, this paper would make a great contribution to our understanding of early dinosaurian evolution

Many thanks; we have tried to do this.

Reviewer #3 (Remarks to the Author):

Dear Editor Nature Communications,

I congratulate the authors for such a well-written and clear MS with well-exposed hypotheses and ideas. The contribution by Bernardi et al. is really important and of deep importance and would result of broad interest on the paleontological community. The MS includes new evidence regarding the origin and early evolution of dinosaurs, based on novel evidence.

I think that the MS should be published after minor review (I attached a PDF including some citations that, I think, authors should include in the final version of their contribution. The statistical analyses are well-founded and are sound.

We have worked through the annotated MS and made all corrections; thee are all listed and commented – not many comments in fact.

Page 2: I think Ezcurra (2010) should be cited here; I think Bonaparte (1982) should be cited here

We are sympathetic to these requests, but we have cited a number of papers already here, and these would be 'desirable, but not essential' in our view. We note we are already slightly above the preferred number of bibliographic references, given the rules of *Nature Communications*.

Page 6: Add here *Powellvenator* (Ezcurra, 2017); here should be also included the Quebrada del Barro fauna (e.g., Martínez et al., 2013, 2015; Apaldetti et al., 2011; Martinez and Apaldetti, 2017)

We include these additional records, but briefly, as the material is not of really good quality, sufficient to name the taxa. Citations are too many to fit in the list, as noted above.

REVIEWERS' COMMENTS:

Reviewer #2 (Remarks to the Author):

G'day,

It was a pleasure editing this manuscript. Your edits and adaptation of suggested alterations has greatly improved this manuscript and feel that it will be a great addition to the scientific literature.

Good luck
Cheers